# Dynamic Changes in Serum Cytokine Profile in Rats with Severe Acute Pancreatitis

**DOI:** 10.3390/medicina59020321

**Published:** 2023-02-09

**Authors:** Rui Zhou, Wangjun Bu, Yudan Fan, Ziwei Du, Jian Zhang, Shu Zhang, Jin Sun, Zongfang Li, Jun Li

**Affiliations:** 1National & Local Joint Engineering Research Center of Biodiagnostics and Biotherapy, The Second Affiliated Hospital of Xi’an Jiaotong University, Xi’an 710004, China; 2Department of General Surgery, The Second Affiliated Hospital of Xi’an Jiaotong University, Xi’an 710004, China; 3Department of Breast and Thyroid Surgery, The Northwest Women’s and Children’s Hospital, Xi’an 710061, China; 4Center for Tumor and Immunology, The Precision Medical Institute, The Second Affiliated Hospital of Xi’an Jiaotong University, Xi’an 710004, China

**Keywords:** severe acute pancreatitis, cytokines, inflammation, disease progression, microarray

## Abstract

*Background and Objectives*: Most published research has only investigated a single timepoint after the onset of severe acute pancreatitis (SAP), meaning that they have been unable to observe the relationship between the dynamic changes in cytokines and SAP progression. In this study, we attempted to reveal the relationship between dynamic changes in cytokine expression levels and SAP disease progression and the relationship between cytokines, using continuous large-scale cytokine detection. *Materials and Methods*: Seventy rats were randomly assigned to control (Con), sham operation (SO) and SAP groups. The SAP group was randomly allocated to five subgroups at 3, 6, 9, 12 and 15 h after the operation. In the SAP group, 5% sodium taurocholate was injected retrograde into the pancreatic bile duct. Animals in the SO group received a similar incision, a turning over of the pancreas. Control animals did not receive any treatment. We observed the survival, ascites fluid amount, pancreatic histopathological scores and serum amylase activity of SAP rats. We used the cytokine microarray to simultaneously detect 90 cytokines and the dynamic changes in one experiment and to analyze the correlation between cytokine expression and disease progression. *Results*: The mortality of SAP rats increased with an increase in time. Serum amylase activity, pancreatic histopathological scores and ascites fluid amount were time-dependent. Compared with normal rats, 69 cytokines in SAP rats were significantly changed for at least one timepoint, and 49 cytokines were significantly changed at different timepoints after SAP induction. The changes in inflammatory cytokines were significantly upregulated at 6 and 9 h and 12 h and then significantly decreased. *Conclusions*: The trend of cytokine expression in SAP rats was not consistent with the disease progression. The cytokine–cytokine receptor interaction and MAPK signal’s dominant cytokines were always highly expressed at various time points over the course of SAP.

## 1. Introduction

Acute pancreatitis (AP) is an acute inflammatory process of the pancreas most commonly caused by bile stones or excessive use of alcohol. Severe acute pancreatitis (SAP) is the severe form of AP and has an increasing incidence in recent years worldwide [1,2]. SAP is a life-threatening disease with high hospital mortality rates of 20–40%. The relatively high mortality of SAP is often linked to multiorgan failure. Despite the considerable social and economic burden associated with SAP, there remains no specific therapy for the disease. Therefore, a detailed study of the pathogenesis of SAP is necessary to identify potential drug targets for transformation.

The pathophysiology of SAP still remains unclear. Inflammatory cytokines play a role in promoting the occurrence and development of SAP. Cytokine cascades are not only involved in the pathogenesis of SAP but also participate in the pathological stages of all types of acute pancreatitis. Once the disease starts, inflammatory pathways are activated, along with repair pathways. A variety of proinflammatory cytokines are released into the blood circulation. Localized inflammatory reactions occur in the injured area, and further development and hyper-proinflammatory response leads to cytokine storm and systemic inflammatory response syndrome (SIRS). Early imbalances in SIRS and subsequent compensatory anti-inflammatory response syndrome (CARS) with susceptibility to sepsis and infections, as well as multiple organ dysfunction syndrome (MODS), are the main causes of SAP morbidity and mortality. Importantly, there is no effective treatment that can change the pathological process of SAP [2,3]. It was reported that the expression of some cytokines can be used as markers for diagnosis and prognosis. However, the research results are still in controversial. The reason for the contradictions is the difference in the response patterns and the release time of different cytokines and the cross-linking relationship between different cytokines. Most of the published research has only investigated a single timepoint after the onset of SAP and is unable to observe the relationship between the dynamic changes in cytokines and the progression in SAP. We attempted to reveal the relationship between dynamic changes in cytokine expression levels and SAP disease progression and the relationship between cytokines through continuous, large-scale cytokine detection.

Cytokines are secreted proteins with a low molecular weight that exert various biological activities in different cells through specific receptors on the cell surface. According to their functions, they can be divided into pro-inflammatory factors, anti-inflammatory factors and inflammatory mediators. Pro-inflammatory factors are represented by activin A, tumor necrosis factor-alpha (TNF-α), interleukin-1 beta (IL-1β), IL-2, IL-6, IL-8 and other factors. Leptin, transforming growth factor-beta 1 (TGF-β1), IL-4, IL-10 and IL-11 are the representative anti-inflammatory factors. Inflammatory mediators mainly include prostaglandins (PGs), thromboxane (TX), leukotrienes (LTs), platelet-activating factor (PAF), nuclear transcription factor-κB (NF-κB) and some other enzymes. Therefore, to observe the dynamic changes in various cytokines as best as possible, we used the cytokine microarray to simultaneously detect 90 cytokines and the dynamic changes in one experiment and to analyze the correlation between cytokine expression and disease progression.

## 2. Materials and Methods

### 2.1. Materials

Male rats (Sprague-Dawley; 260–300 g) were purchased from the Experimental Animal Center of Xi’an Jiaotong University Health Science Center (Xi’an, China). All animals were housed in standard shoebox cages 1 w before surgery and were given free access to standard feed. Animals were housed with a 12:12 h light–dark cycle, given water ad libitum, and the room temperature was controlled at 23 ± 2 °C. Fasting occurred at 12 h and drinking water was given 4 h before surgery. All experiments were performed according to the Institutional Animal Care and Use Committee of Xi’an Jiaotong University Health Science Center. Sodium taurocholate and sodium pentobarbital were from Sigma-Aldrich (St. Louis, MO, USA).

### 2.2. Experimental SAP Model

After fasting overnight, 70 rats were randomly assigned to control (Con, *n* = 10), sham operation (SO, *n* = 10) and SAP (SAP, *n* = 50) groups. The SAP group was randomly allocated to five subgroups for the time points of 3, 6, 9,12 and 15 h after the operation (10 rats in every time point group). For SAP, a sodium taurocholate retrograde injection model was used [4]. Under the sodium pentobarbital anesthetic (50 mg/kg, Sigma-Aldrich), the animals were laparotomized at the midline. The duodenum and the common bile duct were identified. The common biliopancreatic duct was advanced with a catheter and the hepatic duct was closed with a small bulldog clamp to prevent backflow. Sodium taurocholate was solubilized in sterile water at a final concentration of 50 mg/mL and injected retrograde into the common biliopancreatic duct at a dose of 50 mg/kg, given one time (with 0.1 mL/min injection velocity). Then, the injection point was pressed for 3 min and the surgery finished with abdominal stratified closing. Special attention was paid to atraumatic surgical technique. Animals in the SO group received a similar incision but only a turning over of the pancreas. Control animals did not receive any treatment. Animals were sacrificed 3, 6, 9, 12 or 15 h after the injection.

Animals were anesthetized by sodium pentobarbital, and the abdominal cavity was incised and the blood samples and pancreas were harvested for measurement. The development of SAP was confirmed by histological changes and by increased amylase activity in the serum.

### 2.3. Measurement of Mortality in SAP Rats

The mortality of rats was determined by calculating the numbers of surviving and dead rats in every group at the different timepoints.

### 2.4. Detection of Ascites Fluid Amount

Ascites fluid was harvested and weighed. The ascites fluid amount was assayed by the difference in gauze weight, before and after being placed in the abdominal cavity for 5 min.

### 2.5. Histology and Morphometry

For histological evaluation, pancreas tissue was fixed immediately in 4% paraformaldehyde, dehydrated, embedded in paraffin, sectioned at 4 μm and stained with hematoxylin and eosin (H&E). H&E-stained sections of pancreas were evaluated microscopically by a person unaware of the treatment. The level of edema, acinar necrosis, hemorrhage and inflammation indicated the extent of the injury. The extent of injury was calculated by morphometry, as described earlier [5]. Two slides and ten fields/slide were examined for histopathological analysis in each pancreas. Images were captured with a Nikon E100 microscope equipped with a CCD camera.

### 2.6. Electron Microscopy

Ultrastructural examination was performed using a transmission electron microscope according to a previously described protocol [4]. Pancreatic tissue was collected from animals 15 h after SAP induction. The pancreatic tissues were cut into 1 mm cubes and fixed at 4 °C in 2.5% glutaraldehyde overnight. The samples were then immersed in 1% osmium tetroxide at room temperature for 2–3 h and then in uranyl acetate for 2–3 h. The tissues were dehydrated in serial solutions of ethanol and acetone and embedded in epoxy resins. For examination by electron microscopy, 50 nm ultrathin sections were stained with lead citrate and mounted on 200-mesh copper grids. Electron microscopy was performed on a Hitachi H-7650 (Hitachi, Tokyo, Japan) transmission electron microscope.

### 2.7. Measurement of Serum Amylase Activity

The blood was centrifuged at 1800× *g* for 10 min and the serum stored at −80 °C for subsequent analysis. The serum amylase activity was measured using an automatic biochemical analyzer (Roche, Basel, Switzerland). Serum amylase activity was measured to reflect the severity of SAP.

### 2.8. Biochip Array

The profiles of 90 cytokines (inhibin beta A chain (Activin A), adrenocorticotropic hormone (ACTH), adipocyte differentiation-related protein (ADFP), Adiponectin/Acrp30, 5′-AMP-activated protein kinase catalytic subunit alpha-1 (AMPK alpha 1), B7-1/CD80, brain-derived neurotrophic factor (BDNF), beta-Catenin, fibroblast growth factor 2 (basic-FGF), beta-nerve growth factor (beta-NGF), C-C chemokine receptor type 4 (CCR4), CD106, cytokine-induced neutrophil chemoattractant 2 alpha/beta (CINC-2 alpha/beta), CINC-3, ciliary neurotrophic factor (CNTF), CNTF receptor subunit alpha (CNTF R alpha), tyrosine-protein kinase CSK (CSK), C-X-C chemokine receptor type 4 (CXCR4), epidermal growth factor receptor (EGFR), endocrine gland derived vascular endothelial growth factor (EG-VEGF/PK1), E-Selectin, Fas-associating protein with a novel death domain (FADD), tumor necrosis factor superfamily member 6 (Fas Ligand/TNFSF6), tumor necrosis factor ligand superfamily member 6 (Fas/TNFRSF6), fibroblast growth factor binding protein 1 (FGF-BP), follistatin-like-1(FSL1), fractalkine, glialcellline-derived neurotrophic factor family receptor alpha-1 (GFR alpha-1), GFR alpha-2, granulocyte-macrophage colony-stimulating factor (GM-CSF), growth hormone, growth hormone receptor (growth hormone R), hepassocin, intercellular adhesion molecule 1 (ICAM1), intestinal cell kinase (ICK), insulin degrading enzyme, interferon-gamma (IFN-gamma), IL-1α, IL-1 beta, IL-1 R6/IL-1 R rp2, IL-2, IL-3, IL-4, IL-5, IL-6, IL-10, IL-12/IL-23 p40, IL-13, integrin alpha M beta 2, insulin, interferon γ-induced protein 10 (IP-10), leptin(OB), C-X-C chemokine 5 (LIX), L-Selectin/CD62L, monocyte chemotactic protein 1 (MCP-1), macrophage-derived chemokine (MDC), macrophage migration inhibitory factor (MIF), macrophage inflammatory protein 1 alpha (MIP-1 alpha), MIP-2, MIP-3 alpha, matrix metalloproteinase-13 (MMP-13), MMP-2, MMP-8, muscle, skeletal receptor tyrosine-protein kinase (MuSK), Neuropilin-2, nerve growth factor receptor (NGFR), orexin A, osteopotin/SPP1, platelet-derived growth factor-AA (PDGF-AA), prolactin receptor (Prolactin R), receptor for advanced glycation endproducts (RAGE), receptor-associated late transducer/mitogen inducible gene-6 (RALT/MIG-6), resistin-like molecule beta (RELM beta), resistin, tumor necrosis factor ligand superfamily member 15 (TL1A), TGF-β1, TGF-beta2, TGF-beta3, thrombospondin, tunica interna endothelial cell kinase 2 (TIE-2), tissue inhibitor of metalloproteinases 1 (TIMP-1), TIMP-2, TIMP-3, Toll-like receptor 4 (TLR4), TNF-α, tumor necrosis factor-related apoptosis-inducing Ligand (TRAIL), tumor necrosis factor receptor superfamily member 19 (TROY), ubiquitin, vascular endothelial growth factor (VEGF), vascular endothelial growth factor-C (VEGF-C)) were investigated in the serum of SAP rats at different timepoints using a RayBio^®^ L-Series Rat Antibody Array 90 Glass Slide.

The blood samples at different timepoints were centrifuged at 14,000 rpm in a pre-cooled centrifuge for 15 min, and the serum was stored at −80 °C. Briefly, BCA protein quantification was performed on the serum samples to calculate the protein concentration. Briefly, BCA protein quantification was performed on the serum samples to calculate the protein concentration. Subsequently, the antibody coated membrane was placed in a container and incubated for 30 min with 2 mL blocking buffer to block membrane. The blocking buffer was decanted before diluted samples were added to incubate membrane for 1–2 h. The samples were decanted and the membrane was washed with wash buffer. The diluted biotin-conjugated antibodies were added to incubate the membrane for 1–2 h. The membrane was washed and incubated with HRP-conjugated streptavidin at room temperature for 2 h. The membrane was washed and incubated with detection buffer A and detection buffer B for 1 min (KangChen KC-430 kit, Shanghai, China). The membrane was exposed to X-ray film and detected signal using film developer. The image of the film was scanned by a scanner and converted into a grayscale TIFF file. ScanAlyze software was used to convert the dot matrix of grayscale into numeric data, which represent the relative expression levels of cytokines.

### 2.9. Statistical Analysis

Results are expressed as means ± standard deviation (S.D). First, the normality and homogeneity of variance tests were carried out, and data conformed to normal distribution and homogeneity of variance. Statistical analysis was performed using a *t*-test or analysis of variance in SPSS statistical software version 19.0 (SPSS Inc., Chicago, IL, USA). Two-tailed *p*-values are reported.

## 3. Results

### 3.1. Alteration of Mortality

All 20 rats survived in the Con and SO group (mortality rate 0%). The mortality after SAP induction in SAP group was 10% at the 3 h timepoint, 20% at 6 h timepoint, 32% at the 9 h timepoint, 40% at the 12 h timepoint and 48% at the 15 h timepoint. The mortality was increased in SAP group compared with the Con and SO group (Figure 1A, *p* < 0.05).

### 3.2. Alteration of Ascites Fluid Amount

The amount of ascites fluid in the SAP group significantly increased at the 3, 6, 9, 12 and 15 h timepoints compared with the SO group (Figure 1B, *p* < 0.05). More clear ascites appeared in the SAP group at 3 h, with no significant change compared with the SAP ascites rats at 6 h. The ascites of SAP rats at 9 h began to appear as cloudy ascites. After 15 h of SAP, the amount of ascites increased and became bloody with flocs.

### 3.3. Increased Severity of Pancreatic Injury

Control animals showed entirely normal glandular architecture. Apart from a minimal focal edema, SO group also showed entirely normal acinar architecture and no abdominal fluid or pancreatic edema. The SAP group at the 3 h timepoint demonstrated moderate edema and little acinar necrosis and hemorrhage. In the SAP group at the 6 and the 9 timepoints, the histopathologic appearance of pronounced edema and marked interstitial leukocyte infiltration were evident. Twelve and fifteen hours after SAP induction, acinar necrosis was catastrophic and involved entire lobules with extremely patchy distribution. Intense granulocyte infiltration was uniformly present and frequently accompanied by widespread hemorrhage (Figure 1C,E).

Under electron microscopy, a normal pancreas in the control group showed a morphology of normal acinar cells with typical secretory polarized morphology. Many zymogens were concentrated in the apical pole of the cells. The basal region of the acinar cell contained a nucleus surrounded by cisternae of rough endoplasmic reticulum (ER). Many ribosomes were found on the endoplasmic reticulum membrane. Mitochondria were spherical and observed in the basal region of the cell. The acinar cells in SAP 15 h group showed many vacuoles in the apical cytoplasm, endoplasmic reticulum dilatation and swollen mitochondria (Figure 2).

### 3.4. Alteration of Serum Amylase Activity

The serum amylase activity in the SAP group significantly increased at the 3, 6, 9, 12 and 15 h timepoints compared with the SO group and showed a time dependence (Figure 1D, *p* < 0.05).

### 3.5. Alteration of Cytokine Profiles in SAP

To assess the differences in cytokine profiles in SAP rats at different time points, the broad profiles of cytokines in the serum were measured via a biochip array. We detected the expression of 90 cytokines in the serum of SAP rats at 6 h, 9 h, 12 h and 15 h. There were significant differences in the cytokine profiles between the SAP group and normal group. The cytokine profiles of SAP at the 6 h timepoint and 9 h timepoint were similar, while those of 12 h and 15 h were more similar. Compared with normal rats, 69 cytokines in SAP rats were significantly changed for at least one timepoint, and 49 cytokines were significantly changed at least two time points after SAP induction (Figure 3).

The serum amylase activity, pancreatic histopathological score and ascites fluid amount at each timepoint were time-dependent, along with the SAP progression, but the alterations in serum cytokines were not. The 53 cytokines were upregulated at the 9 h timepoint compared with the 6 h time point. Of the 53 cytokines, 31 were upregulated by at least 1.3-fold. Compared with the 9 h timepoint, 60 cytokines were downregulated by at least 1.3-fold at the 12 h timepoint. For the pro-inflammatory and anti-inflammatory cytokines that are most associated with inflammation, most cytokines showed an obvious downward trend at the 12 h timepoint and then were significantly upregulated at the 15 h timepoint. This trend of cytokine profiles was contrary to the pathological course of SAP rats. The expression trend in cytokines with similar functions was basically the same, and the upward and downward trends of pro-inflammatory factors and anti-inflammatory factors were also the same (Figure 4).

Compared with the Con group, the SAP group exhibited significant elevations in the levels of BDNF, Leptin, β-Catenin, ACTH, B7-1/CD80, Fractalkine, β-NGF, ADFP, TIE-2, IntegrinαMβ2, TGF-β2, GFR α1, Fas/TNFRSF6 and TGF-β3 at all four timepoints (Figure 5A). ActivinA, BDNF, beta-catenin, CCR4, Fractalkine and GM-CSF were upregulated more than twice at each timepoint in the SAP group. ACTH, B7-1/CD80, GFR-α-1, TGF-β3, TIE2 were upregulated more than five times and leptin was upregulated more than 200 times higher in the SAP group. To gain insight into the probable biological functions of the differentially expressed cytokines, we classified these cytokines by Gene Ontology (GO) and Kyoto Encyclopedia of Genes and Genomes (KEGG) pathway analyses. As shown in Figure 5, GO analysis indicated that 14 differentially expressed cytokines were enriched in terms of single-organism processes, biological regulation, binding and molecular transducer activity. The KEGG pathway analysis showed that the differentially expressed cytokines were mainly involved in cytokine–cytokine receptor interactions and the MAPK signaling pathway (Figure 5B,C).

## 4. Discussion

The role of cytokines in the development and progression of pancreatic inflammatory diseases has been intensively studied over recent decades. Cytokines can regulate each other and form a complex network. They are the message transmitters of the immune system, as well as the bridge between nervous, endocrine and other systems and the immune system, and play a central role in the occurrence, development and recovery of SAP [6,7]. A severe inflammatory response can aggravate overcompensation and the response of the immune system and increase systemic infection risk [8]. During the occurrence and development of SAP, a large number of cytokines are activated, released and involved, and each cytokine participates in different ways and at different timepoints and dosages, forming a three-dimensional, complex and interrelated cytokine network. Previous studies on cytokines in the process of SAP development mainly focused on a single cytokine at a specific time point, and the serum cytokine level can reflect the severity of SAP. At present, a variety of single cytokines, such as IL-1β, IL-6, IL-8, IL-10, TNF-α, CXCR4, MIF, leukotriene B4 (LTB4) and PAF, were used to evaluate SAP and are believed to be related to the occurrence and development of SAP [9,10,11,12,13,14].

In this study, we investigated the circulating cytokine profiles of SAP rats at different timepoints and compared the correlation between cytokine expression and disease progression. The expression of cytokines in the serum of SAP rats significantly increased, including at the early stage (6, 9 h), compared with the control group. However, after 12 h, the proinflammatory cytokines TNF-α, Activin A, FSL-1, GM-CSF, MCP-1, MIP-2, RALT/MIG-6), TNFRSF6 and anti-inflammatory cytokines basic-FGF, CCR4, CNTF, IL-10, MIP-3, MuSK, TL1A, TGF-β1, β-catenin and TIMP-1 were significantly decreased compared with the Con group.

Among these differentially expressed cytokines, TNF-α was an early SAP event, and TNFRSF6 and FSL1 were associated with NLRP3 activation and apoptosis [11,15,16,17,18]. TNF-α participates in the activation of NK and T lymphocytes. MCP-1 and GM-CSF play an immunomodulatory role, activating monocytes and neutrophils, and are associated with promoting the activation of NF-κB at the early stage of inflammatory diseases [19,20,21,22]. Activin A plays a role, promoting the transformation of monocytes into macrophages and regulating the expression of prostaglandins and cytokines [23]. Inhibiting activin A reduces the severity of acute pancreatitis and local immune cell infiltration [24]. MIP-2 plays a major role in mediating the neutrophilic inflammatory response and activating inflammatory cells [25]. Basic-FGF can induce Erk1/2 phosphorylation in nociceptive neurons and induces Erk1/2-dependent mechanical hyperalgesia [26]. CNTF plays an anti-inflammatory role by inhibiting TNF-α production. IL-10 participates in the inhibition of pro-inflammatory cytokine secretion and the secretion of nitric oxide (NO) from monocytes and macrophages. IL-10 gene polymorphism contributes to the development of acute pancreatitis. IL-1α and IL-1β are important regulators of innate and adaptive immunity [27]. CCR4 exerts its immunoregulation effect through CCR4 ^+^ Treg cells and CCR4 ^+^ Th2 cells [28]. β-catenin is a key molecule in the Wnt/β-catenin signaling pathway; there was evidence that activation of Wnt/β-catenin could inhibit inflammatory cytokine release and pancreatic and intestinal damages of AP rats [29]. MuSK reduces the production of active lipids and free calcium levels; TGF-β1 acts by promoting cell transformation, angiogenesis and the proliferation of the extracellular matrix; TIMP-1 acts by inhibiting matrix metalloproteinases (MMPs) activity, controls major aspects of cell biology, and its deregulation contributes to various diseases [30]. We can observe that the expression trend of most cytokines is not linearly upregulated or downregulated with the disease progression. The expression of cytokines increased or decreased before the 12 h timepoints, and the expression trend reversed after 12 h. We can speculate that the cytokine storm does not break out at a single time point, and its triggering mechanism remains to be investigated.

Among these early increases, decreases at 12 h and increases at 15 h, leptin was reported in the literature as an indicator of SAP risk. Plasma leptin levels were increased in both animal models and patients with SAP progression. Injecting animals with exogenous leptin can also reduce the severity of SAP. Among these cytokines, only EGFR reaches its peak value in the later stage of SAP progression, which we speculate may be related to the maximum pancreatic tissue damage and excessive activation of EGFR signal in the later stage of SAP. Further studies of this factor may find its value in indicating poor SAP prognosis.

A previous study showed that the cytokines cascade exacerbates the imbalance in anti-inflammatory and pro-inflammatory cytokines, ultimately leading to SIRS and even MODS, which are important factors in the development and poor prognosis of SAP. In our study, we found that the upregulated and downregulated trends of both pro-inflammatory and anti-inflammatory factors were highly consistent in SAP rat serum. How did the quantitative imbalance between pro-inflammatory and anti-inflammatory cytokines occur, leading to the imbalance in their effects? This mechanism needs to be clarified by further experiments and clinical studies.

The response modes and release times of different cytokines were different, and there was a cross-linking regulatory relationship between different cytokines. A hierarchical cluster analysis and KEGG pathway analysis showed that the cytokine–cytokine receptor interaction and MAPK signal dominant factor were always highly expressed during SAP progress. Subsequent studies should focus on studying the signaling pathway and finding strategies to suppress the early inflammatory response and prevent subsequent SIRS and organ failure.

Although this study did not find that the expression trend of a single cytokine was completely consistent with the pathological progress of SAP, the investigation on dynamic changes in cytokines made us realize that the cytokine expression of SAP presents its own characteristics. Future studies on dynamic changes in cytokines of SAP should not only be used to explore the prediction of the severity and prognosis of SAP. The more important implication should be to observe the dynamic changes in cytokines by interfering with the cytokine signaling pathway and to explore more optimized SAP therapy.

## 5. Conclusions

In summary, in this study, we showed that the trend of cytokine expression in SAP rats was not consistent with the disease progression. The cytokine–cytokine receptor interaction and MAPK signal dominant cytokines were always highly expressed at various timepoints over the course of SAP.

## Figures and Tables

**Figure 1 medicina-59-00321-f001:**
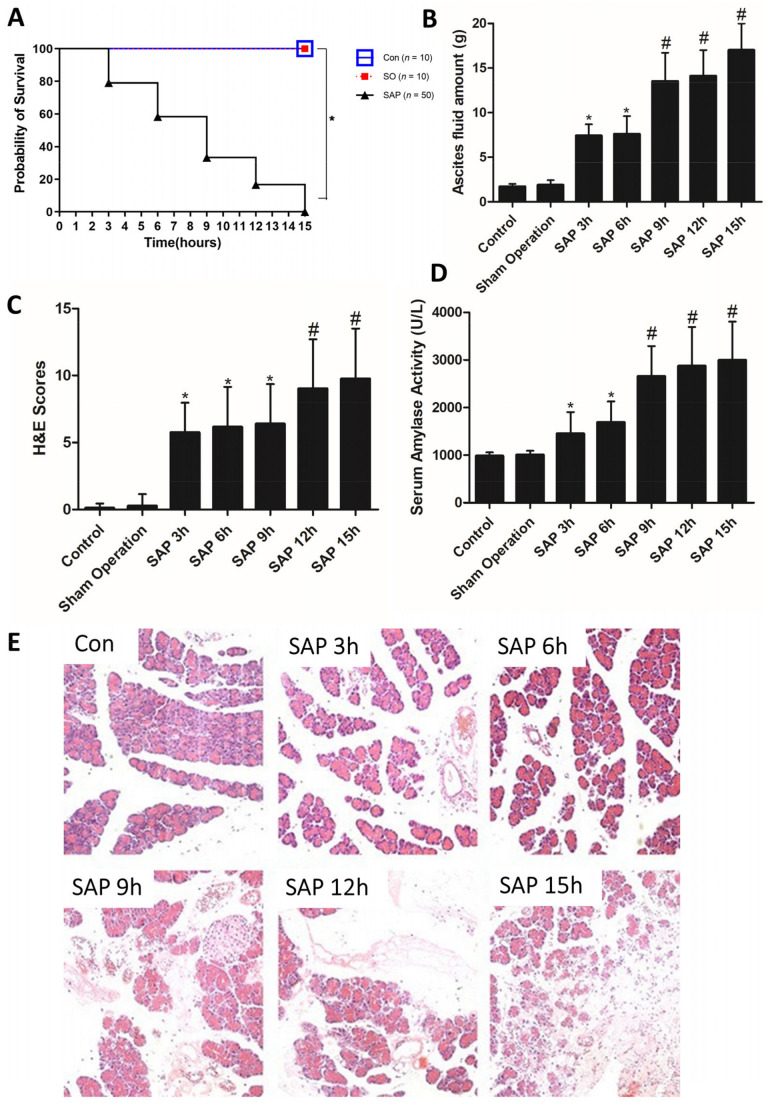
Mortality, pancreatic injury and amylase activity that showed significant differences between groups. (**A**) The mortality of severe acute pancreatitis (SAP) rats at 3, 6, 9, 12 and 15 h. (**B**) Ascites fluid amount at 3, 6, 9, 12 and 15 h of SAP rats. (**C**) Histopathological scores were evaluated complying with the standardized scoring system described in the Materials and Methods. (**D**) Serum amylase activity in different groups. (**E**) Histopathological structure of pancreas (H&E, ×100). * *p* < 0.05 versus the sham operation (SO) group; # *p* < 0.05 versus the SAP 3 h group; Con: Control.

**Figure 2 medicina-59-00321-f002:**
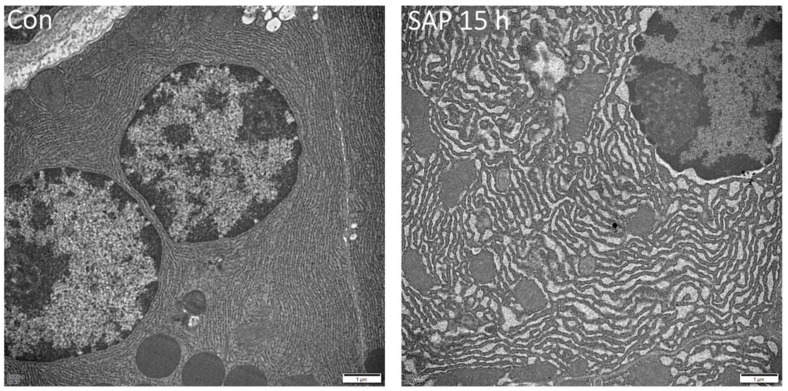
Ultrastructure of pancreas. Control (Con) group showed entirely normal acinar architecture. severe acute pancreatitis (SAP) group showed vacuolization, dilatated endoplasmic reticulum (ER) and cytolysis (×15,000).

**Figure 3 medicina-59-00321-f003:**
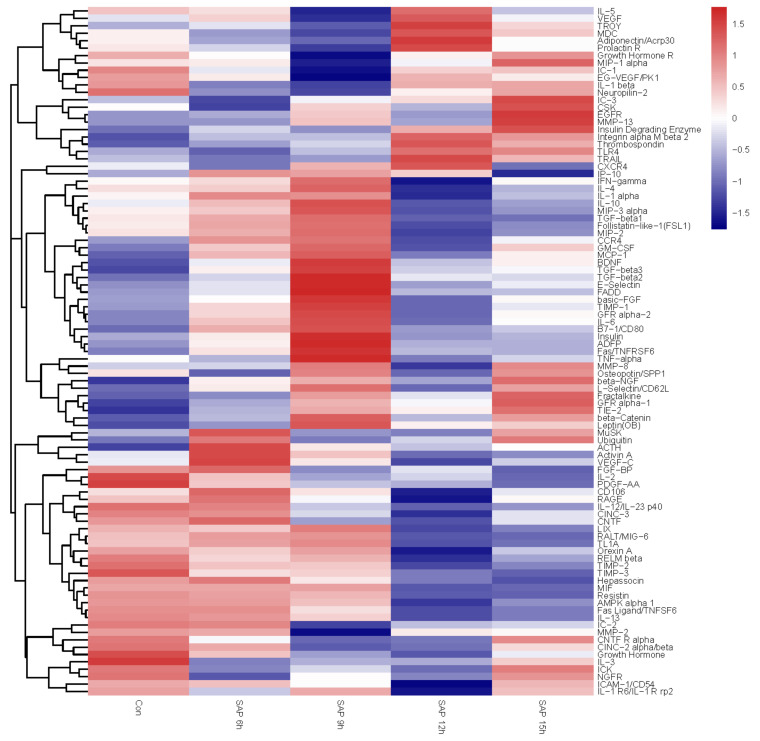
Hierarchical cluster analysis of the 90 cytokines, differentially expressed in rats of different groups. Firebrick: high expression; navy: low expression; white: moderate expression.

**Figure 4 medicina-59-00321-f004:**
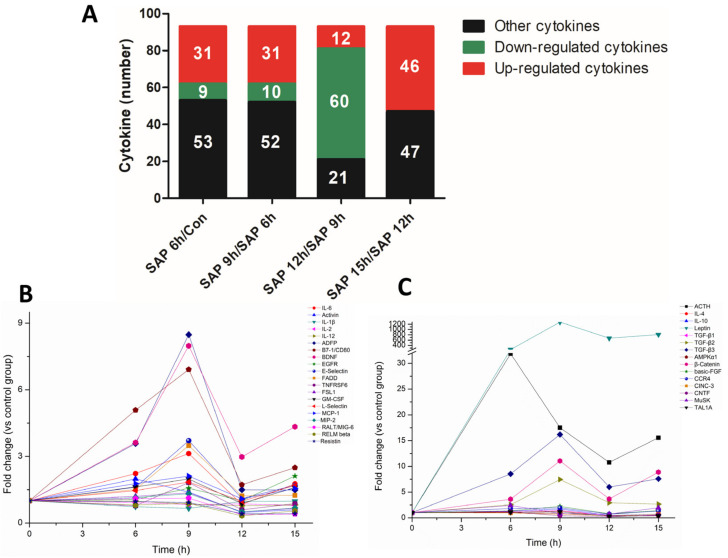
The trend of cytokine expression in SAP rats was not consistent with the disease progression. (**A**) Upregulated (fold change > 1.3) and downregulated (fold change < 1.3) cytokine numbers in different groups compared to the previous time point. (**B**) Alterations in proinflammatory cytokines with time. (**C**) Alterations in anti-inflammatory cytokines with time.

**Figure 5 medicina-59-00321-f005:**
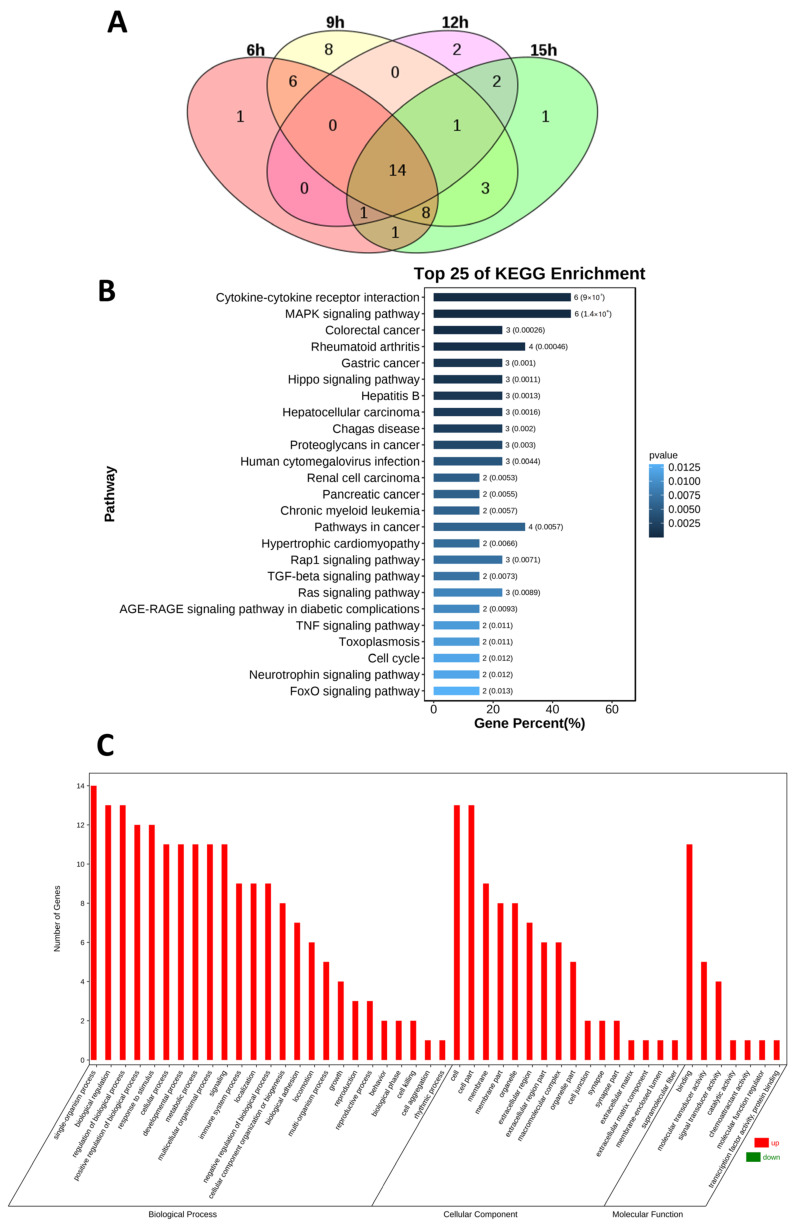
Gene Ontology (GO) and Kyoto Encyclopedia of Genes and Genomes (KEGG) pathway analyses. (**A**) Compared to control group, 14 cytokines were upregulated at four time points. The numbers in the figure represent the number of cytokines. (**B**) Cytokines were classified by GO analysis according to their biological process, molecular function and cellular component. (**C**) The KEGG pathway analysis showed that the differentially expressed cytokines were mainly involved in cytokine–cytokine receptor interactions and MAPK signaling pathway.

## Data Availability

The data presented in this study is available by request from the corresponding author.

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
