# Peer review of "Dynamic Changes in Serum Cytokine Profile in Rats with Severe Acute Pancreatitis"

_medicina, 2023, doi:10.3390/medicina59020321_

Round 1

Reviewer 1 Report

This article is a basic research article that try to observe the relationship between the dynamic changes in cytokines and the progression in SAP. The trend of cytokine expression in SAP rats was not consistent with the disease progression. These observations do not contradict the literature.  They have no value for predicting gravity of the disease.

1. I would like the statistical part that supports the statements to be better described, possibly with values not only with p.

2. I would like newer and more relevant references

Reviewer 2 Report

This is an interesting paper was evaluated numerous markers (amylase, cytokines) and histological changes in rats for severe acute pancreatitis (SAP). This article seems to become the important basic science for uncovering the SAP.

#1

The authors concluded,” The trend of cytokine expression in SAP rats was not consistent with the disease progression”.

However, some cytokines (ADFP, BDNF, B7-1) are elevated during the disease onset and disappear later, while others (Leptin, ACTH) are elevated later in the disease.

Some cytokines that increase later in the disease may be used to assess the risk SAP.

In addition, some cytokines might be discovered the reason for increasing in each timing in the future.

Please discuss the relationship between some cytokines and the timing, especially early and late.

#2

Please consider revising the title considering question #1.

#3

The order of presentation in Figure 3 is 9h>Con>6h>12h>15h. Is this correct?

#4

P-value should be unified in capital letter or lower-case letter (P or p).

Round 2

Reviewer 2 Report

I think that the authors well responded to my comments.

Thank you.